# Loss in the Crowd: Hidden Breakthroughs in Language Model Training

## Abstract

The training loss curves of a neural network are typically smooth. Any visible discontinuities draw attention as discrete conceptual breakthroughs, while the rest of training is less carefully studied. In this work we hypothesize that similar breakthroughs actually occur frequently throughout training, though their presence is obscured when monitoring the aggregate train loss. To find these hidden transitions, we introduce POLCA, a method for decomposing changes in loss along an arbitrary basis of the low rank training subspace. We use our method to identify clusters of samples that exhibit similar changes in loss through training, disaggregating the overall loss into that of smaller groups of conceptually similar datapoints. We validate our method on synthetic arithmetic and natural language, showing that POLCA recovers clusters which represent easily interpretable breakthroughs in the model's capabilities whose existence would otherwise be lost in the crowd.

## 1 Introduction

Recent work on phase transitions during training has characterized the underlying development of structures and mechanisms. These sudden drops in loss reveal the formation of induction heads (Olsson et al., 2022b), syntactic attention structure (Chen et al., 2024a), hierarchical bias (Murty et al., 2023), and many other conceptual breakthroughs (McGrath et al., 2022; Lovering et al., 2022; Power et al., 2022; Abbe et al., 2021). However, the loss curve as a whole remains stubbornly smooth. Phase transitions and momentary conceptual breakthroughs are therefore treated as isolated curiosities; the vast majority of training time is seen as predictable. This work will show that in fact, the model undergoes many hidden abrupt breakthroughs that are concealed by aggregating all data and all dimensions into a single loss curve.

We decompose the loss in two different ways simultaneously to find hidden breakthroughs. First, we decompose the aggregate loss into loss over individual examples or homogeneous subsets of data. By clustering the loss curves of individual examples, we identify subsets of data that experience synchronized changes in loss stability, implying that they rely on the same conceptual breakthroughs. However, any individual example might benefit from multiple conceptual breakthroughs; in such cases, the example may undergo multiple changes that are synchronized with different subsets of training data. In order to disentangle these breakthroughs, we must instead find different mechanisms or internal changes that affect the loss curve for a given example. This requirement brings us to the second axis of decomposition.

Because we need to disentangle multiple relevant concepts, our second decomposition transforms changes in loss into a collection of responses to movement in specific directions during training. By analyzing these loss curves along specific bases, we identify conceptual breakthroughs that rely on a particular direction of movement. The latter analysis permits further granularity in clustering data, as final performance on an individual example may rely on multiple conceptual breakthroughs, each corresponding to a particular linear direction in training.

In summary:

- By clustering datapoints based on their loss changes during training, we discover that concepts are learned at specific **breakthrough** times. Using changes in datapoint loss to measure stability, we show that smooth aggregated loss curves can conceal momentary inflections in datapoint loss, a scenario we describe as **breakthrough elision**.

- We introduce a modified form of Loss Change Allocation (Lan et al., 2020) called Projection Oriented Loss Change Allocation (POLCA) to measure changes in loss due to parameter adjustments in arbitrary directions during training. Using POLCA, we extend our cluster analysis to identify conceptual breakthroughs that occur in a restricted gradient subspace. We use this breakthrough clustering analysis to identify specific concepts that are learned at a breakthrough, in both synthetic and natural language settings.

## 2 BACKGROUND

**What can we learn from transitions in stability?**   Previous work has extensively documented phase transitions in the stability and sharpness of the loss surface. Jastrzębski et al. (2020) point to a clear phase transition in the gradient variance early in training, and Ma et al. (2022) show that such behavior could arise due to the existence of multiple different scales of loss.

**Why disaggregate the aggregate loss?**   Individual samples often exhibit changes in loss that are out of line with the monotonic average trend (Xia et al., 2023; Rosenfeld & Risteski, 2024). In full-batch gradient descent, Cohen et al. (2022) identified non-monotonicity arising from oscillation about the maximum Hessian eigenvector. Rosenfeld & Risteski (2024) gave evidence that these oscillations occur across different axes for different samples, and they highlighted human-interpretable semantic features of the data as a likely cause. We hypothesize that movement in these separate directions signals the model's acquisition of distinct capabilities (i.e. "skills" (Arora & Goyal, 2023; Chen et al., 2024b)). To test this hypothesis, and to better identify the semantic meaning of each of these directions, we propose to decompose this instability—defined as the magnitude of oscillation—according to a basis derived from the full loss Hessian at various training checkpoints.

**Why decompose the overall train loss?**   Similar to the quantization model of parameter scaling of Michaud et al. (2024), we aim to cluster datapoints according to the skills they rely on. However, our POLCA decomposition also addresses what Michaud et al. (2024) call *polygenic* scaling effects—samples which combine multiple skills and therefore exhibit breakthroughs at multiple scales. If we assume that a specific skill is enabled by movement along a particular skill basis vector, then the loss change attributed to movement along the skill basis vector will stabilize at the moment the skill is acquired—for every sample that requires that skill. In this manner, the sample transitions from early to late dynamics through a basis-specific loss phase transition. In other words, by monitoring changes in directions corresponding to specific skills, we support the speculation of Nanda et al. (2023) that "*phase transitions are everywhere.*"

**Why is linear decomposition sufficient?**   In practice, a conceptual breakthrough might not occur in a single direction that persists throughout training. However, there is an abundance of evidence that the linear bases of the low rank training subspace (Gur-Ari et al., 2018) are conceptually meaningful. In the late stages of training, linear interpolation between a pair of checkpoints yields a convex path in the loss space (Frankle et al., 2020). Although independently finetuned models with similar generalization heuristics are also linearly connected, interpolations from a nonlinear connection between a model pair with unmatched heuristics fail to generalize with either heuristic (Juneja et al., 2023, ref Appendix D). These observations suggest that linear decomposition should give good results, and our experiments show that the resulting clusters are interpretable in practice.

## 3 METHODS

The key to our approach is the separate consideration of how each individual example's **datapoint loss** changes throughout training. We contrast this individualized metric with the evaluation of in-distribution performance simultaneously across the entire training or validation set, which we call the **aggregated loss**. Using the datapoint loss, we can cluster individual examples on the basis of their loss $L(w_t)$, change in loss $L(w_t) - L(w_{t-1})$, or magnitude of change $|L(w_t) - L(w_{t-1})|$ during training.

### 3.1 PROJECTION ORIENTED LOSS CHANGE ALLOCATION (POLCA)

Our next objective is to decompose the loss itself into specific directions in the weight space, motivated by several considerations: First, while we have moved from an aggregated loss metric to a more granular datapoint loss metric, we are still only considering breakthroughs that are general enough to be perceived in loss curves. Second, an individual datapoint may benefit from a variety of conceptual breakthroughs, but will not be clustered on the breakthroughs individually. Finally, once we have identified a subset of the data as benefiting from a particular conceptual breakthrough, decomposing into individual weight directions allows us to locate where in the weights the breakthrough occurs and to thereby identify the mechanism involved.

Next we break this loss down by directional movement during training, allowing us to discover breakthroughs that are specific to a given direction. Our procedure, Projection Oriented Loss Change Allocation (POLCA), comprises two steps: first, the selection of the basis, followed by the decomposition of the loss according to that basis.

#### 3.1.1 FINDING THE BASIS

---

**Algorithm 1** Finding the POLCA basis

---

**input:** Training set $X$, Model checkpoints $\{\theta_t\}_{t=1}^{T}$.
$B \leftarrow \emptyset \in \mathbb{R}^{d \times 0}$.
**for** $t = 1 \ldots T$ **do**
    $\Pi_{\perp} \leftarrow I - B(B^{\top}B)^{-1}B^{\top}$
    $\mathcal{H} \leftarrow \nabla_{\theta}^2 \mathcal{L}(X, \theta)$.
    Define $B^{+} \in \mathbb{R}^{d \times k}$ as the top $k$ eigenvectors of $\Pi_{\perp}\mathcal{H}$ (e.g., via the Lanczos method).
    $B \leftarrow [B, B^{+}]$.
**end for**
**return** $B$

---

We focus on a restricted subspace when decomposing the loss, selecting the basis of this subspace from the maximum eigenvectors of the Hessian matrix. We posit this basis to be interpretable because each basis vector expresses a high gradient covariance and therefore represents a potential decision boundary.

This basis is constructed as follows. Given $T$ intermediate checkpoints throughout training of a model with weights in $\mathbb{R}^d$ and a number $k$ of eigenvectors to compute at each checkpoint, we seek a low rank $Tk$-dimensional subspace which captures most of the movement during optimization (Gur-Ari et al., 2018). We construct this basis iteratively, starting with $B = \emptyset$: at each checkpoint $t$ we project the model's loss Hessian onto the nullspace of $B \in \mathbb{R}^{d \times (t-1)k}$. We then identify the top $k$ eigenvectors of the resulting projection and append these to $B$, expanding the dimension. We compute the eigenvectors using Hessian-vector products to avoid explicitly constructing the full Hessian matrix. The resulting basis is designed to include directions of highest curvature at each checkpoint so that it will capture synchronized loss behavior throughout training. Note that the very top eigenvectors are likely to reflect oscillation, rather than conceptually meaningful movement (Song et al., 2024), but as we continue to add to the low rank basis, we include more directions of stable movement as well.

#### 3.1.2 DECOMPOSING THE LOSS

To decompose the loss along our basis, we propose a modified version of Loss Change Allocation (LCA; Lan et al., 2020). LCA is an interpretability tool for analyzing changes in aggregated loss on dataset $X$ between two checkpoints. The output of LCA is the empirical loss change between a pair of checkpoints which can be attributed to the motion of each individual weight unit. Given two consecutive checkpoints with parameters $\theta_t$ and $\theta_{t+1}$, LCA reformulates the change in loss as its first-order Taylor approximation, a sum of components which each attribute some loss change to the

movement of a single parameter unit $\theta^{(j)}$:

$$L(X; \theta_{t+1}) - L(X; \theta_t) \approx \sum_{j=0}^{d} (\nabla_\theta L(X; \theta_t))^{(j)} (\theta_{t+1}^{(j)} - \theta_t^{(j)}) \tag{1}$$

$$= \sum_{j=0}^{d} LCA(X; \theta_t^{(j)}) \tag{2}$$

The POLCA decomposition differs from LCA in three key ways. First, we do not restrict each direction to correspond to a single unit $\theta^{(j)}$, instead permitting an *arbitrary basis* vector $b \in B$ to replace the axis-aligned basis vectors in LCA; we project onto this basis vector using the dot product $\langle b, \cdot \rangle$. Second, we are interested in changes in the loss on each individual example $x \in X$, not the entire dataset $X$. These first two modifications provide the following reformulation of LCA.

$$L(X; \theta_{t+1}) - L(X; \theta_t) = \sum_{x \in X} L(x; \theta_{t+1}) - L(x; \theta_t) \tag{3}$$

$$\approx \sum_{x \in X} \sum_{b \in B} \langle b, \nabla_\theta L(x; \theta_t) \rangle \langle b, \theta_{t+1} - \theta_t \rangle \tag{4}$$

The third key difference is that we must use a second order approximation because this basis is constructed explicitly from the Hessian eigenvectors. To understand why this choice of basis requires a second order approximation, recall that each basis vector $b$ is an eigenvector of the Hessian matrix $\mathcal{H}_{t'}(X)$ at some timestep $t'$, where $b$ is chosen because it has the largest eigenvalue $\lambda_{t'}(X, b)$ over the whole dataset. If we assume that the top eigenvectors of the aggregate Hessian maintain high curvature at other points in training and on individual datapoints, then the scaling factor in the second order Taylor term will be very large even at the datapoint level. Limiting the approximation to only the first order term gives poor guarantees on error, as the second order term could be expected to dominate. We find that empirically, the difference between the first and second order values is small (Appendix F), but compute the second-order approximation to achieve a better estimate.

Exact computation of the second order term would be intractable, requiring computation of the top eigenvalues/vectors for each individual datapoint $x$. Instead, we can approximate it by substituting the true eigenvalue, denoted $\lambda_t(X, b) := b^\top \mathcal{H}_t(X) b$, with the curvature of the individual loss in the direction $b$, i.e. $\lambda_t(x, b) = b^\top \mathcal{H}_t(x) b$. If the aggregate Hessian eigenvector $b$ is close to the span of the top eigenvectors of the datapoint-specific Hessian for $x$, this provides a reasonable estimate while reducing calculation to a single Hessian-vector product per eigenvector. We therefore approximate the basis projection of the datapoint Hessian $h(x, b, \theta_t)$ as detailed in Appendix A.

$$h(x, b, \theta_t) = \frac{\lambda_t(x, b)}{2} \langle \theta_{t+1} - \theta_t, b \rangle^2$$

$$\approx \frac{\lambda_t(X, b)}{2} \cdot \langle \theta_{t+1} - \theta_t, b \rangle^2 \times \frac{\langle L(x; \theta_{t+1}) - L(x; \theta_t), b \rangle}{\langle L(X; \theta_{t+1}) - L(X; \theta_t), b \rangle} \tag{5}$$

$$= \tilde{h}(x, b, \theta_t)$$

Equipped with this second order approximation of the basis projection of the datapoint Hessian, we account for the high curvature and possible domination of the second order term over the first order term by modifying Equation 4 into the second order Taylor expansion using the approximation from Equation 5.

$$L(X; \theta_{t+1}) - L(X; \theta_t) \approx \sum_{x \in X} \sum_{b \in B} \langle b, \nabla_\theta L(x; \theta_t) \rangle \langle b, \theta_{t+1} - \theta_t \rangle + \tilde{h}(x, b, \theta_t) \tag{6}$$

$$= \sum_{x \in X} \sum_{b \in B} POLCA(x, b; \theta_t) \tag{7}$$

## 4 ARITHMETIC LANGUAGE MODELING

We validate our method in a synthetic setting and find that breakthrough clustering can, in fact, reveal concepts of discrete and natural kinds within the data, even when those kinds are not discoverable by clustering directly on loss curves.

| OUTPUT TOKEN | MAXIMUM LOSS CLUSTER CARRY HOMOGENEITY | POLCA VECTORS WITH >90% CARRY HOMOGENEITY CLUSTERS |
|---|---|---|
| 0 | 0.18 | [0] |
| 1 | 1.00 | [0–19] |
| 2 | 0.31 | [0–6, 11, 12] |
| 3 | 0.35 | [4] |
| 4 | 0.43 | [4, 10, 12, 16] |
| 5 | 0.64 | [4, 5, 12, 15, 17] |
| 6 | 0.45 | [8, 15] |
| 7 | 0.53 | [0, 1, 4, 6, 8] |
| 8 | 0.49 | [2, 6, 7, 8, 13, 14] |
| 9 | 0.41 | [10, 12] |

Table 1: Clustering recovers information about the arithmetic carry skill. For each output token type, we report the maximum fraction of token instances requiring the carry skill in the loss breakthrough clustering and the set of POLCA vectors with breakthrough clusters containing at least 90% carry skill instances. Clustering based on loss only results in a homogeneous cluster for token 1 whereas POLCA recovers homogeneous carry clusters for all output tokens.

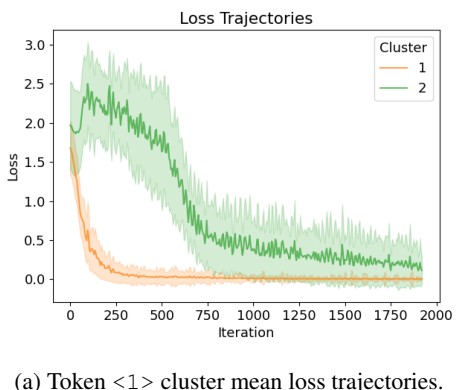

(a) Token <1> cluster mean loss trajectories.

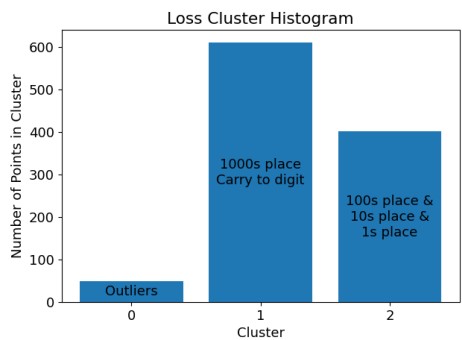

(b) Token <1> cluster labels and their counts.

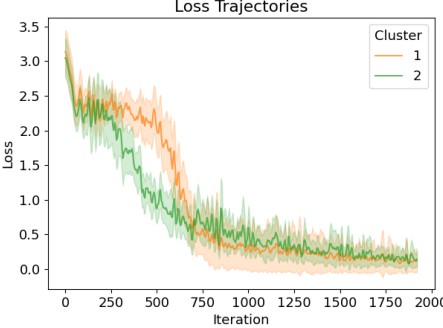

(c) Token <3> cluster mean loss trajectories.

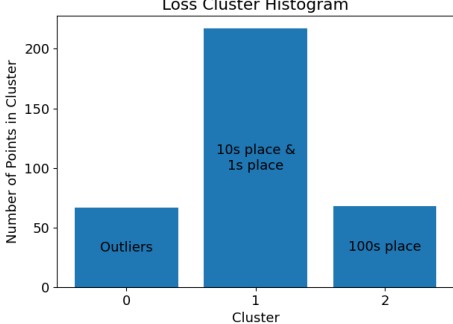

(d) Token <3> cluster labels and their counts.

Figure 1: Clusters on the arithmetic task associated with output tokens <1> (1a-1b) and <3> (1c-1d). Clusters are labeled with a set of digit places if over 90% of the token instances in the cluster belong to the set of digit places. Clusters are labeled with carry or no carry if over 90% of the token instances in the cluster require an arithmetic carry to the target token.

## 4.1 THE DATA

Our synthetic experiments use data from the arithmetic addition setting in Chen et al. (2024b), where the model is trained to compute the sum of two 3-digit numbers. This setting has 4 skills corresponding to each of the digits in the output sum. Note that the digit in the 1000s place is always

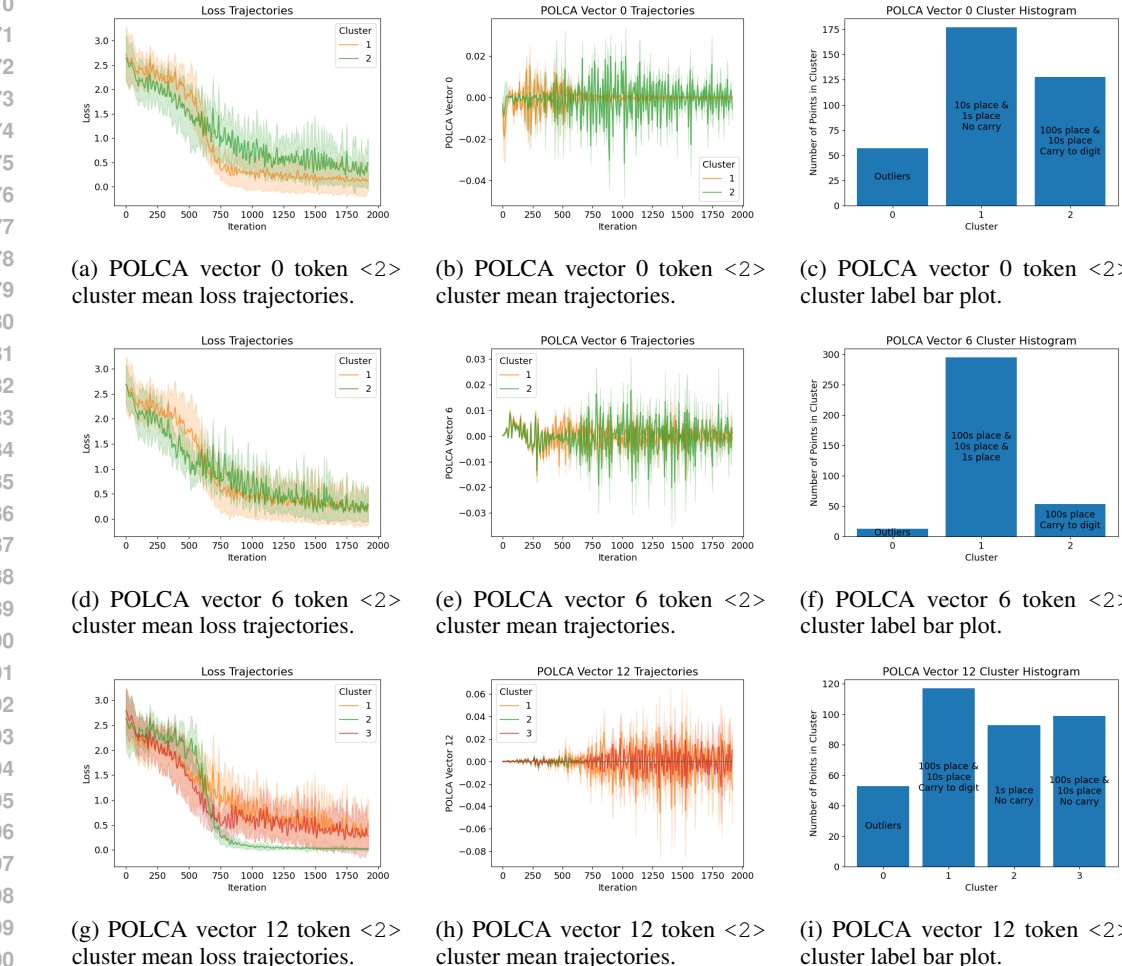

(a) POLCA vector 0 token <2> cluster mean loss trajectories.

(b) POLCA vector 0 token <2> cluster mean trajectories.

(c) POLCA vector 0 token <2> cluster label bar plot.

(d) POLCA vector 6 token <2> cluster mean loss trajectories.

(e) POLCA vector 6 token <2> cluster mean trajectories.

(f) POLCA vector 6 token <2> cluster label bar plot.

(g) POLCA vector 12 token <2> cluster mean loss trajectories.

(h) POLCA vector 12 token <2> cluster mean trajectories.

(i) POLCA vector 12 token <2> cluster label bar plot.

Figure 2: Arithmetic language modeling breakthrough clustering case study. For output token <2>, we report average cluster loss curves, POLCA trajectories, and cluster label bar plots for breakthrough clustering on POLCA vector 0 (2a-2c), vector 6 (2d-2f), and vector 12 (2g-2i). Bar plots are labeled with a given set of digits if over 90% of the token instances in the cluster belong to the set and are labeled with carry (no carry) if over 90% of the token instances in the cluster belong to the carry (or no carry) category. The breakthrough clustering on POLCA trajectories (the vector of POLCA values for a token instances throughout training) recovers different versions of the carrying concept for different basis vectors.

a <0> or <1> token since the two input summants are 3 digits long. As shown in Appendix Figure 3 and Chen et al. (2024b), the skills corresponding to the digits have different loss curves, so the digit skill categories may be recoverable by clustering the loss curves. We also consider an additional skill: arithmetic carries to the output token. The digit is a simple skill that can be clustered using the overall loss, whereas the carries represent skills that are not necessarily clear from the overall loss (see Appendix Figure 4), potentially requiring the POLCA decomposition.

**Experimental setup**   We train a 2-layer transformer model with embedding dimension 512, 4 attention heads per layer, and an MLP dimension of 2048. We choose this model size to align with prior work Olsson et al. (2022a) and to maximize the granularity at which we can feasibly compute the POLCA values. For a validation set with 1250 data points and 5000 output tokens, we compute the loss and POLCA values for each token at intervals of 5 iterations throughout training. We choose the training steps between each POLCA computation to achieve as fine-grained analysis as possible without exploding the compute time. We compute the POLCA basis using the eigenvectors of the Hessian estimated using a 1250 data point sample of the training set as detailed in Algorithm 1. We

compute one new basis vector every 100 iterations for a total of 30 basis vectors since we did not observe significantly different results from adding more basis vectors in our preliminary experiments.

We then analyze breakthrough clustering on the loss and POLCA trajectories in 4.2 and 4.3. For each possible output token [<0>, <1>, <2>, <3>, <4>, <5>, <6>, <7>, <8>, <9>], we cluster the loss trajectories for all of the instances of that token in the validation set. Here, a trajectory is the vector of loss (or POLCA) values computed for a specific token instance throughout training. By clustering these trajectories, we can discover skills learned for each output token. We use Hierarchical Density-Based Spatial Clustering of Applications with Noise (HDBSCAN) Campello et al. (2013) as the clustering algorithm since we are interested in discovering clusters with various densities (i.e. with curves that are close together or farther apart) and since it can identify outliers. Outlier selection is very important in this setting, as we seek to identify which data points are not relevant to a skill being learned in a specific direction. We set the minimum cluster size to be at least 20% of the total number of trajectories to ensure that the clusters are significant. This results in 2-3 clusters for each output token. The HDBSCAN outliers are labeled and shown as cluster 0 in the bar plots but excluded from the remaining analysis in 4.2 and 4.3.

## 4.2 RECOVERING CONCEPTS FROM THE EXACT LOSS

In our clustering experiments on arithmetic addition skills, we first consider whether directional decomposition is necessary for identifying individual concepts. To this end, we cluster solely on the exact per-token loss curves for successive timesteps, rather than estimating the loss decomposed along a low rank basis. In Figure 1, which shows a sample of the loss cluster labels, we do find that it is possible to recover, to a substantial degree, the digit skill by clustering only on the loss, likely because the digits have very different loss trajectories.

As shown in Table 1 (and Appendix Figure 5), clustering on the loss alone does not recover homogeneous clusters with respect to the carrying skill except for the output token <1>. For <1>, the carrying skill is likely recovered because of the digit skill, as the carry cluster for <1> consists of token instances in the 1000s place (Figure 1). For the 1000s place, the output can only be <1> if there is a carry to it due to the construction of the three digit arithmetic task. We will demonstrate a clear improvement in the recovery of the carry skill and the interpretability of clusters after POLCA decomposition.

## 4.3 RECOVERING CONCEPTS WITH POLCA

Due to the shortcomings of clustering solely on the loss, we instead cluster on the loss changes decomposed by POLCA, separately considering each basis vector. The POLCA value for a given token and basis vector represents the loss change attributed to movement along that vector. We find that for all possible output tokens, there is at least one POLCA vector with homogeneous clusters corresponding to carrying skills (Table 1). Thus, POLCA is able to recover complex skills such as carries that are challenging to cluster using the exact loss changes alone. *We conclude that arithmetic carries rely on breakthroughs along specific dimensions during training, but these breakthroughs may be elided in the exact loss curve computed across all dimensions.*

Figure 2 shows an example of the POLCA breakthrough clusters for output token <2> (see Appendix E for clusters for other tokens). Note that the construction of the POLCA basis (Algorithm 1) assigns larger indices to the eigenvectors added from later points in training, which are also likely to have smaller eigenvalues across training. These later basis vectors tend to cluster on compositions of the digit and carry skills rather than solely the carry skill. For example, they separate the digit that is being carried to (Figure 2f) or split the no carry tokens into multiple clusters (Figure 2i). This finding indicates that these lower eigenvalue directions are important for more fine-grained versions of skills than the higher eigenvalue directions earlier in training. The trend of fine-grained skills corresponding to lower eigenvalue directions is especially clear from POLCA vector 12, where the LCA value is extremely small for all of the clusters in the beginning of training (Figure 2h). In addition, for all of the POLCA basis vectors in Figure 2, the average magnitude of the LCA value for the carry cluster increases sharply at some point in training. As the POLCA vector number increases, this sharp increase occurs at later iterations. This indicates that there are phase transitions at different iterations for the carry skill along different directions, which are obscured when only looking at the loss.

Thus, the clusters for different POLCA vectors recover skills that are difficult to find by clustering only on the loss and can be used to understand which directions are important for learning a specific skill and when these directions emerge as top eigenvectors in the Hessian. As a result, we have shown that breakthrough clustering on the POLCA vectors can be used to find which directions complex skills are learned along and to better understand how they are learned.

## 5 NATURAL LANGUAGE MODELING

We apply our approach to a real-world causal language modeling task and show that POLCA breakthrough clustering recovers interpretable conceptual skills in the natural language setting.

### 5.1 EXPERIMENTAL SETUP

For the language modeling setting, we use the Wikipedia dataset (Wikimedia Foundation, 2022), which consists of the clean English language Wikipedia dump from March 2022. We use the same setup that we validate in the arithmetic case. We train a 2-layer transformer model with embedding dimension 512, 4 attention heads, and an MLP dimension of 2048. We compute the loss and POLCA values for each token in a validation set with 6350 output tokens at intervals of 100 iterations throughout training. We compute the POLCA basis using the eigenvectors of the Hessian estimated using a 1000 data point sample of the training set as detailed in Algorithm 1 with $k = 1$. We compute a new basis vector every 200 iterations.

We then analyze breakthrough clustering on the loss and POLCA trajectories in 5.2. Similarly to the arithmetic setting, for each output token type that we are interested in analyzing and each POLCA vector, we perform clustering on all of the POLCA trajectories corresponding to instances of that token. We use HDBSCAN clustering to cluster the instances of that token and ignore trajectories marked as outliers by HDBSCAN. We set the minimum cluster size to be at least 20% of the total number of trajectories so that the clusters are of significant size.

### 5.2 RECOVERING NATURAL LANGUAGE CONCEPTS WITH POLCA

We apply the clustering approach validated in 4.3 to the natural language setting and use POLCA to analyze the learned skills related to specific output tokens. As there are too many unique output tokens to analyze individually, we focus on a case study of the frequently occurring output tokens < and> and <,>. We find relevant skills by manually inspecting the context (or 20 preceding tokens) of instances of these two tokens with high magnitude LCA values. We label each instance of these tokens with the skills in their context. We then perform breakthrough clustering for each token and POLCA vector combination. We label each cluster with a given skill if over 85% of the top 10% of trajectories closest to the centroid of the cluster represent instances of the output token that involve the skill.

We report a selection of clusters corresponding to specific skills in Table 2. Using breakthrough clustering on POLCA vectors, we find subsets of instances of each output token that correspond to skills such as predicting the token < and> in a list (POLCA vector 0 cluster 2), predicting the token <,> after a number (POLCA vector 0 cluster 2), , and predicting the token < and> after a comma (POLCA vector 7 cluster 1). In other words, it appears that concepts related to lists are learned along specific directions at particular times, allowing them to be clustered separately from predictions of the same token under different conditions. We note that in contrast to the POLCA analysis, out of the two tokens analyzed, clustering on the loss trajectories only results in one homogeneous cluster corresponding to the list skill for the < and> token.

The skill labels show which subsets of POLCA vectors the given skill is learned along for each output token. We note that many of the skills (such as predicting the token after a capitalized word and after a newline) occur in clusters for the same POLCA vector for both output tokens, indicating that similar skills are learned for different tokens along similar sets of basis vectors. As a result, we have shown that POLCA breakthrough clustering can be used to better understand how different skills are learned during training in the natural language setting.

Table 2: Natural language clusters recovered using POLCA. Using POLCA, we find multiple contexts for each target token that decompose onto different basis vectors. We report the top 3 contexts (20 tokens before the output token instance) closest to each cluster centroid. The reported loss cluster is the only homogeneous cluster for any of the tokens analyzed.

| TOKEN | VECTOR | CLUSTER | SKILL | CONTEXTS CLOSEST TO CLUSTER CENTROID |
|---|---|---|---|---|
| < and> | Loss | 1 | List | ' metric system, having the unit symbol kg.  It is a widely used measure in science, engineering and', ' figure in GermanBroke may refer to:\n\nArts, entertainment, and media\n\nFilm and', ', 1848March 28, 1905), courtesy name Gongdu (), was a Chinese official, scholar, and' |
| < and> | 0 | 2 | List | '.  It had six other UK offices in Croydon, York, Birmingham, Crawley, Swanley and', ' the unit symbol kg.  It is a widely used measure in science, engineering and commerce worldwide, and', ' School in Jonesville, South Carolina, where he was all-conference in football, basketball, and' |
| <,> | 0 | 2 | Token after number | 'Chamber of Deputies in 1873, and President of the Senate in 1892.  He became Vice President in 1894,', ' at "De la Brooke Manor" in St.  Mary\'s County, Maryland on March 11, 1785,', ' president when Luis Cordero left office.\n\nHe was Minister of Finance in 1873, 1883,' |
| < and> | 4 | 1 | Token after capitalized word | ' Telugu and two films in Hindi.\n\nSince 2001, he has won 6 Filmfare Awards South and', ' gecko, a lizard in the family Gekkonidae.  The species is endemic to India and', ' primarily located on the western half of the East Frisian peninsula, to the east of West Frisia and' |
| <,> | 4 | 2 | Token after capitalized word | ' The anchor stores are JCPenney, Shoe Carnival, Barnes & Noble, Planet Fitness,', 'running newspaper strip Funky Winkerbean.\n\nCareer\nBorn in Akron, Ohio,', ' Catholic Church) in Kerala, India.\n\nBorn in Thanneermukkom, Kerala,' |
| < and> | 7 | 1 | Token after comma | '1895 and 1 September 1895.\n\nSalazar was Presidents of the Chamber of Deputies in 1873, and', ' As well as the granite-built house, the complex includes numerous outbuildings and greenhouses, and', ' later in the same place.  His important contributions to China made him a recognised figure of his time, and' |
| < and> | 7 | 2 | Token after capitalized word and newline | 'Telugu and two films in Hindi.\n\nSince 2001, he has won 6 Filmfare Awards South and', ' as president of the Poetry Society (UK) and Poetry Editor at The New Yorker.\n\nLife and', '\n Thomas WalterTrevor Ricardo Nelson, MBE (born 7 January 1964) is a British DJ and' |
| <,> | 7 | 2 | Token after newline | ', and a park.\n\nKultaranta's original owner was the businessman AlfredIn chemistry,', 'ordero left office.\n\nHe was Minister of Finance in 1873, 1883, 1884–1887,', ' from 2006 until 2008.\n\nEarly life\nMcKillop was born in Dreghorn,' |

## 6 CONCLUSIONS AND FUTURE WORK

In this work, we have introduced POLCA, a method to compute changes in loss due to parameter changes in arbitrary basis vector directions during training. POLCA decomposes the loss on two levels: individual datapoints and specific directions in the weight space. We show that this decomposition can be used to find breakthroughs in training that are hidden by the aggregate loss. To do so, we perform clustering on POLCA trajectories and demonstrate that this recovers complex skills that are obscured when analyzing the loss alone. We use POLCA to identify and analyze skills learned at breakthroughs during training in synthetic and natural language data.

**Future work** Our method of constructing a basis is inspired by the existing literature on training in restricted subspaces. However, a limitation of this approach is that the top eigenvectors of the Hessian, like the axis-aligned basis, are likely to represent many concepts in superposition. Therefore, we expect that there are gradient directions that represent interpretable concepts more cleanly but

may exhibit dependencies due to superposition. In addition, the top Hessian eigenvectors result in a basis with changes in oscillation magnitude rather than movement in POLCA value, so we may obtain more useful clusters by finding directions that have less oscillation. As a result, future work exploring different bases for POLCA computation may yield improved results. Alternatively, as many of the skills (such as carries in the synthetic setting and predicting the token after a capitalized word in the natural language setting) are recovered when clustering on the POLCA trajectories for more than one basis vector, another useful approach could be to combine these polysemantic orthogonal basis vectors to cluster on their shared concepts.

A limitation of the experiments in this work is that they are computed with small models. We use small models so that we can have very fine-grained checkpoints for POLCA computation. One extension of this work is to explore extending it to larger models, which may require using a basis that is less computationally expensive to compute than Hessian eigenvectors.

A separate direction of future work involves using insights from POLCA to improve other methods. For example, as POLCA elucidates how different skills are learned in training, insights from POLCA analysis could be used to improve optimization. POLCA analysis could also be combined with model editing methods to better remove abilities learned by the model.

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

## A  DERIVATION OF APPROXIMATE SECOND ORDER TERM

We can approximate the difference between the gradient at time $t$ and $t+1$ as

$$
\begin{align}
g_{t+1}(X) - g_t(X) &\approx \mathcal{H}_t(X)(\theta_{t+1} - \theta_t) \tag{8}\\
\langle g_{t+1}(X) - g_t(X), b \rangle &\approx b^\top \mathcal{H}_t(X) b \langle b, \theta_{t+1} - \theta_t \rangle \tag{9}\\
&= \lambda_t(X) \langle b, \theta_{t+1} - \theta_t \rangle \tag{10}
\end{align}
$$

If we assume $b$ to also be an eigenvector of the datapoint Hessians $\mathcal{H}'_t(x)$, we can apply a similar argument for the gradient on the datapoint level.

$$
\langle g'_{t+1}(x) - g'_t(x), b \rangle \approx b^\top \mathcal{H}'_t(x) b \langle b, \theta_{t+1} - \theta_t \rangle \tag{11}
$$

Note that the assumption above (of matching Hessians between data points and their aggregate) is unlikely to be correct. If this assumption is violated, then the scaling factor in the following second order Taylor term will be minuscule on the datapoint level. In practice, we have found that the second order term has limited impact at the datapoint level (see Appendix F), but we nonetheless use it to improve our approximation. Then we may approximate it as:

$$
\begin{align}
\frac{\langle g'_{t+1}(x) - g'_t(x), b \rangle}{\langle g_{t+1}(X) - g_t(X), b \rangle} &\approx \frac{b^\top \mathcal{H}'_t(x) b \langle b, \theta_{t+1} - \theta_t \rangle}{\lambda_t(X, b) \langle b, \theta_{t+1} - \theta_t \rangle} \tag{12}\\
\frac{\langle g'_{t+1}(x) - g'_t(x), b \rangle}{\langle g_{t+1}(X) - g_t(X), b \rangle} &\approx \frac{\langle h'_t(x), b \rangle \langle b, \theta_{t+1} - \theta_t \rangle}{\lambda_t(X, b) \langle b, \theta_{t+1} - \theta_t \rangle} \tag{13}\\
\lambda_t(X, b) \frac{\langle g'_{t+1}(x) - g'_t(x), b \rangle}{\langle g_{t+1}(X) - g_t(X), b \rangle} &\approx \langle h'_t(x), b \rangle \tag{14}
\end{align}
$$

## B  HYPERPARAMETERS

In the tables below, we provide the hyperparameters used during training of the models in the synthetic arithmetic and language modeling settings.

Table 3: Hyperparameters for training the synthetic arithmetic model

| HYPERPARAMETER | VALUE |
|---|---|
| Number of Parameters | 6323210 |
| Iterations | 3000 |
| Epochs | 1 |
| Batch size | 64 |
| Number of training tokens | 768000 |
| Optimizer | AdamW |
| Learning rate | 1e-5 |
| Weight decay | 0.1 |
| Betas | $(0.9, 0.95)$ |
| LR Schedule | $\min(i/100, 1.0)$ |

Table 4: Hyperparameters for training the natural language model

| HYPERPARAMETER | VALUE |
| --- | --- |
| Number of Parameters | 37122353 |
| Iterations | 2500 |
| Epochs | 1 |
| Batch size | 64 |
| Number of training tokens | 20480000 |
| Optimizer | AdamW |
| Learning rate | 1e-5 |
| Weight decay | 0.1 |
| Betas | $(0.9, 0.95)$ |
| LR Schedule | $\min(i/100, 1.0)$ |

## C  UNDECOMPOSED TRAJECTORIES FOR THE DIGIT AND CARRY SKILLS

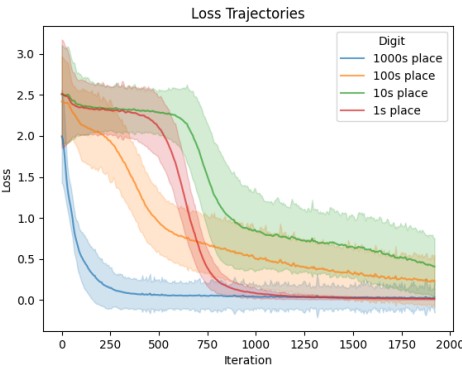

Figure 3: Mean and standard deviation of the loss trajectories for each digit.

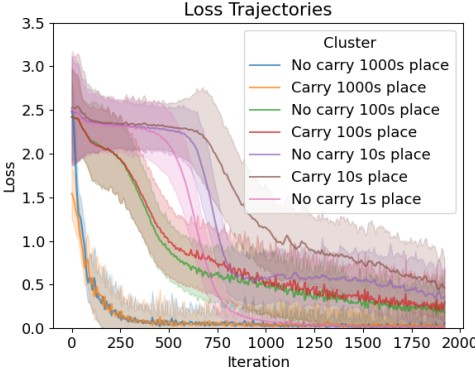

Figure 4: Mean and standard deviation of the loss trajectories for each digit and carry combination.

# D    COMPARISON OF BREAKTHROUGH CLUSTERING FOR THE CARRIES SKILL

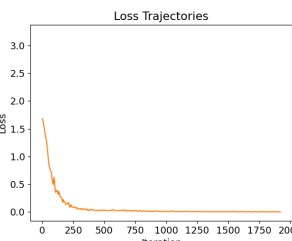 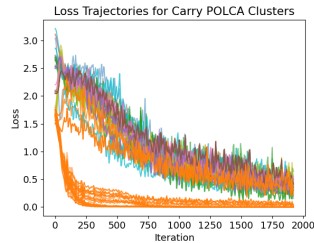 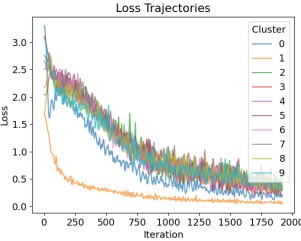

(a) Homogeneous carry loss clusters colored by output token.

(b) Homogeneous carry POLCA clusters colored by output token.

(c) Ground truth mean loss curves clustered by carry and output token.

Figure 5: Average cluster loss curves for different breakthrough clustering methods on the skill-it addition dataset, and for the ground truth subsets that correspond to each cluster's dominant set of skills. Using POLCA and visualizing the clusters with over 90% carries, we find clusters corresponding to the carrying skill for each output token, which are challenging to recover using only the loss.

# E ADDITIONAL ARITHMETIC CLUSTER EXAMPLES

(a) POLCA vector 4 token <4> cluster mean loss trajectories.

(b) POLCA vector 4 token <4> cluster mean trajectories.

(c) POLCA vector 4 token <4> cluster label bar plot.

(d) POLCA vector 10 token <4> cluster mean loss trajectories.

(e) POLCA vector 10 token <4> cluster mean trajectories.

(f) POLCA vector 10 token <4> cluster label bar plot.

(g) POLCA vector 16 token <4> cluster mean loss trajectories.

(h) POLCA vector 16 token <4> cluster mean trajectories.

(i) POLCA vector 16 token <4> cluster label bar plot.

Figure 6: Arithmetic language modeling breakthrough clustering case study. For output token <4>, we report average cluster loss curves, POLCA trajectories, and cluster label bar plots for breakthrough clustering on POLCA vector 4, vector 10, and vector 16. Bar plots are labeled with a given set of digits if over 90% of the token instances in the cluster belong to the set and are labeled with carry (or no carry) if over 90% of the token instances in the cluster belong to the carry (or no carry) category. The breakthrough clustering recovers different versions of the carrying concept for different basis vectors.

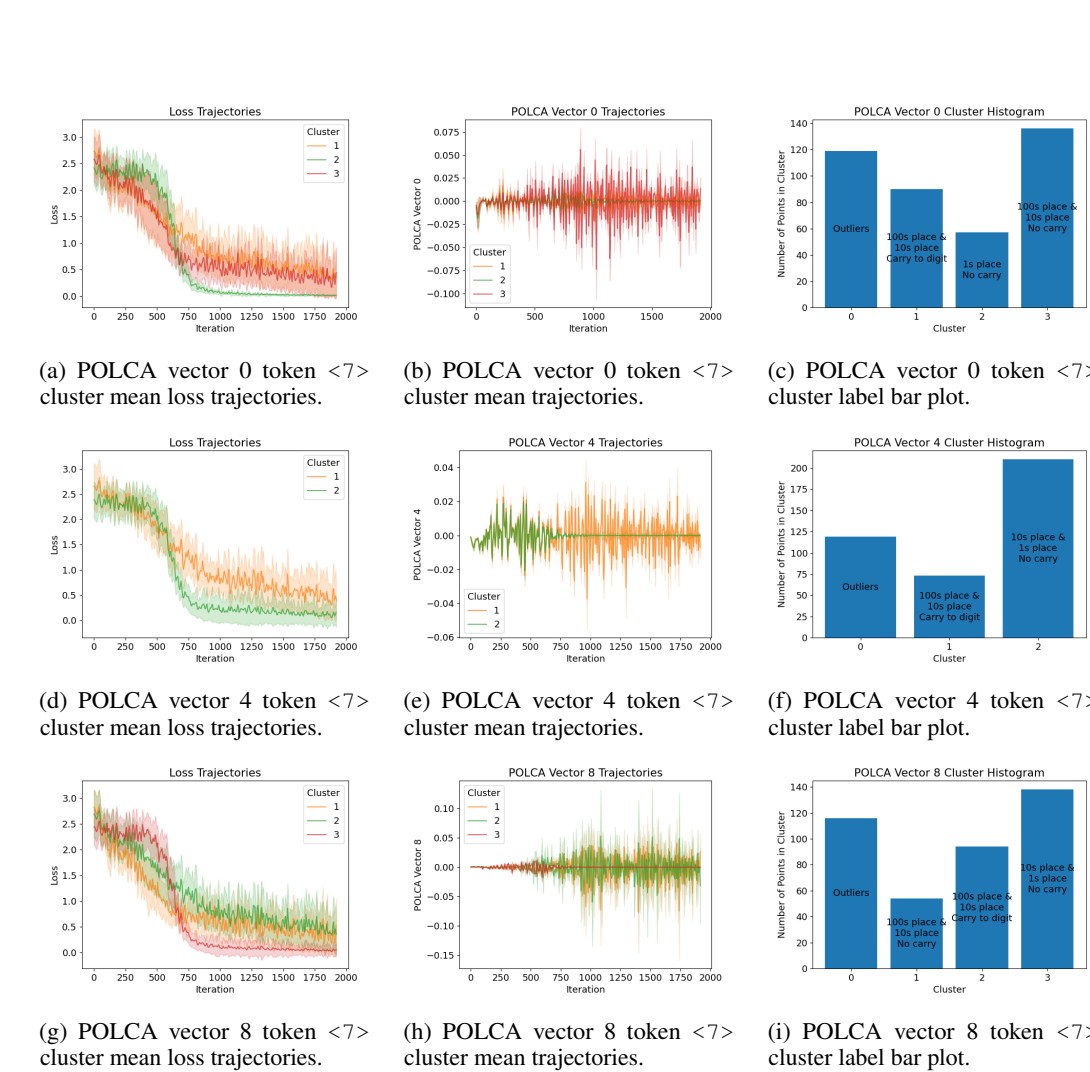

(a) POLCA vector 0 token <7> cluster mean loss trajectories.

(b) POLCA vector 0 token <7> cluster mean trajectories.

(c) POLCA vector 0 token <7> cluster label bar plot.

(d) POLCA vector 4 token <7> cluster mean loss trajectories.

(e) POLCA vector 4 token <7> cluster mean trajectories.

(f) POLCA vector 4 token <7> cluster label bar plot.

(g) POLCA vector 8 token <7> cluster mean loss trajectories.

(h) POLCA vector 8 token <7> cluster mean trajectories.

(i) POLCA vector 8 token <7> cluster label bar plot.

Figure 7: Arithmetic language modeling breakthrough clustering case study. For output token <7>, we report average cluster loss curves, POLCA trajectories, and cluster label bar plots for breakthrough clustering on POLCA vector 0, vector 4, and vector 8. Bar plots are labeled with a given set of digits if over 90% of the token instances in the cluster belong to the set and are labeled with carry (or no carry) if over 90% of the token instances in the cluster belong to the carry (or no carry) category. The breakthrough clustering recovers different versions of the carrying concept for different basis vectors.

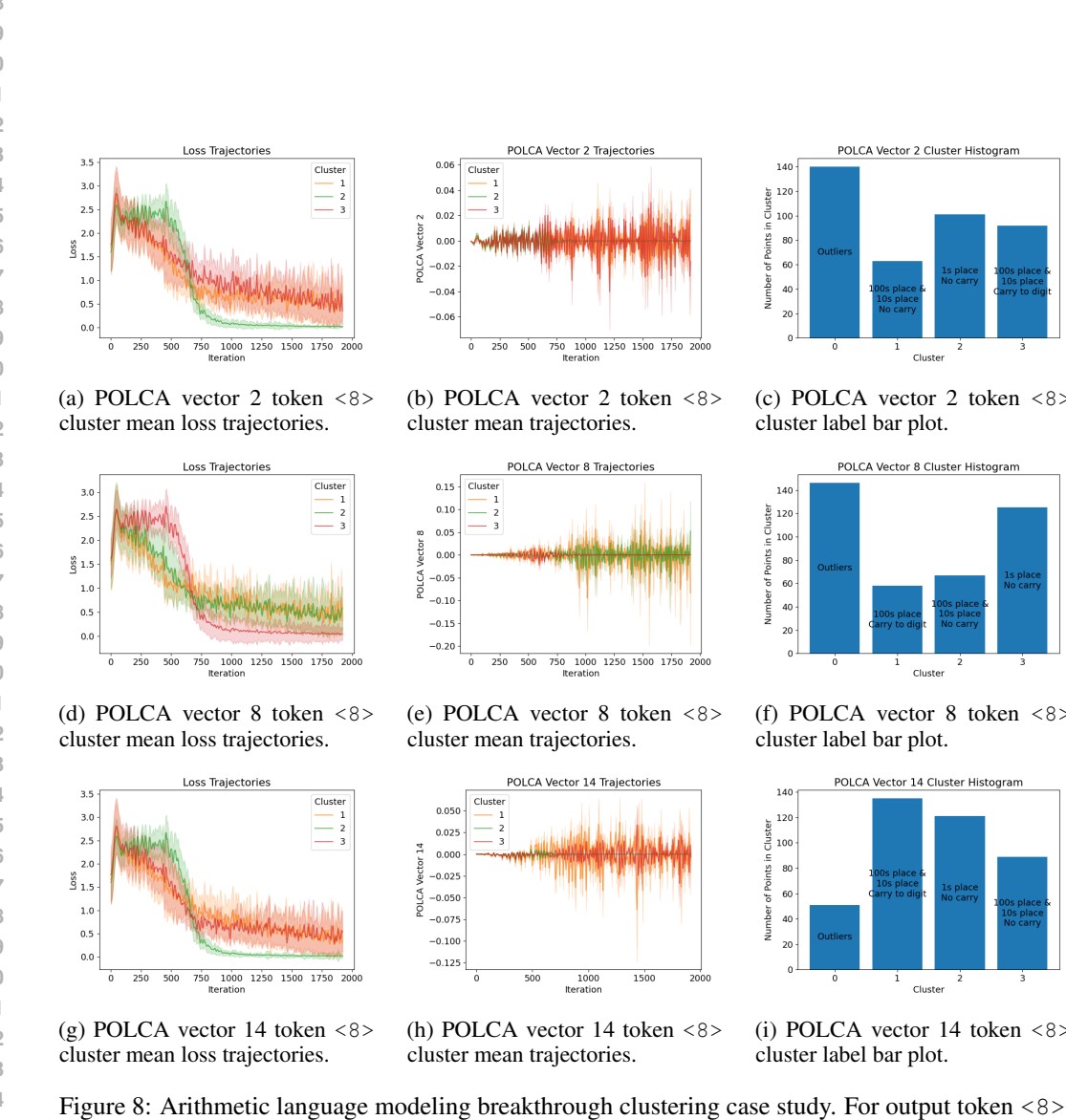

(a) POLCA vector 2 token <8> cluster mean loss trajectories.

(b) POLCA vector 2 token <8> cluster mean trajectories.

(c) POLCA vector 2 token <8> cluster label bar plot.

(d) POLCA vector 8 token <8> cluster mean loss trajectories.

(e) POLCA vector 8 token <8> cluster mean trajectories.

(f) POLCA vector 8 token <8> cluster label bar plot.

(g) POLCA vector 14 token <8> cluster mean loss trajectories.

(h) POLCA vector 14 token <8> cluster mean trajectories.

(i) POLCA vector 14 token <8> cluster label bar plot.

Figure 8: Arithmetic language modeling breakthrough clustering case study. For output token <8>, we report average cluster loss curves, POLCA trajectories, and cluster label bar plots for breakthrough clustering on POLCA vector 2, vector 8, and vector 14. Bar plots are labeled with a given set of digits if over 90% of the token instances in the cluster belong to the set and are labeled with carry (or no carry) if over 90% of the token instances in the cluster belong to the carry (or no carry) category. The breakthrough clustering recovers different versions of the carrying concept for different basis vectors.

## F SECOND VERSUS FIRST ORDER POLCA APPROXIMATION

Table 5: Empirical comparison of second and first order POLCA values. For the arithmetic setting, we compute the average cosine similarity and L2 norm of the difference between the second (Eq 5) and first (Eq 4) order POLCA trajectory vectors. The first and second order approximations of the POLCA trajectories are very similar on average.

| COSINE SIMILARITY | L2 NORM |
|---|---|
| 5.4891 E-4 | 0.99987 |

## G CHOICE OF BASIS

In this section, we analyze the choice of POLCA basis. To do so, we compute the cosine similarity between the original basis constructed using the top Hessian eigenvectors and a variety of other choices of basis vectors.

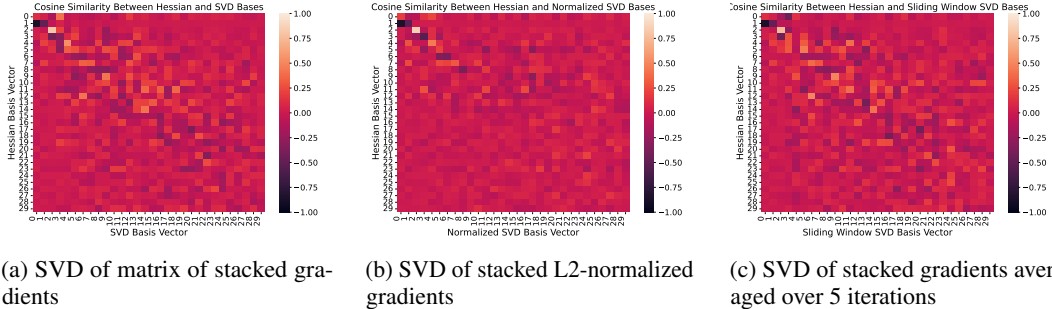

(a) SVD of matrix of stacked gradients

(b) SVD of stacked L2-normalized gradients

(c) SVD of stacked gradients averaged over 5 iterations

Figure 9: Pairwise cosine similarity between the vectors in the Hessian eigenbasis and vectors in various bases constructed using the singular value decomposition (SVD) of the matrix of stacked individual gradients. The Hessian basis vectors are similar to the basis vectors computed using the SVD in all cases, especially for the lower vector numbers, which tend to have higher eigenvalues. However, there is lower similarity between the vectors with the higher numbers (and lower eigenvalues), especially in the normalized case.

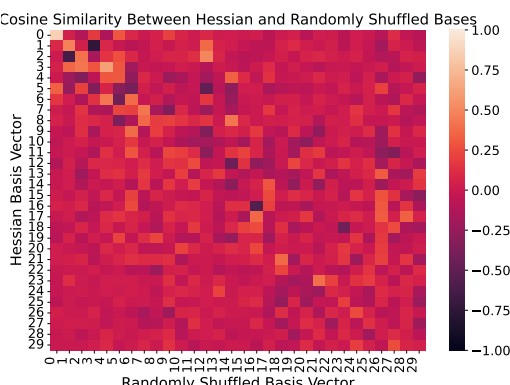

Figure 10: Comparison of Hessian basis to randomly shuffled Hessian basis. We report the cosine similarity between the Hessian basis vectors and a set of Hessian basis vectors computed by randomly shuffling the order of the checkpoints used to construct the basis. The vectors are re-ordered after computation so that the vector number corresponds to the original ordering. The corresponding eigenvectors have relatively high cosine similarity, especially for smaller vector numbers (which tend to have larger eigenvalues).

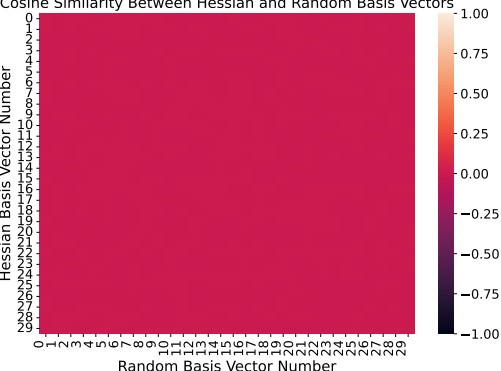

Figure 11: Comparison of Hessian basis to random orthonormal basis. The cosine similarity is very low for all of the vectors in the random orthonormal basis and much lower than the highest values in Figures 9 and 10.

