# OpenReview forum: "Loss in the Crowd: Hidden Breakthroughs in Language Model Training"
_ICLR.cc/2025/Conference — Submitted to ICLR 2025_

### Official Review · Reviewer_Zbn4 · 2024-10-20

**Soundness:** 2
**Presentation:** 3
**Contribution:** 2
**Rating:** 5
**Confidence:** 3

**Summary:**

The paper proposes an interpretability technique (POLCA) that decomposes changes in training loss along both individual data points and orthogonal directions in parameter space, obtained as the largest eigenvectors of the loss Hessian. This decomposition is then used to cluster data points by their corresponding training loss curves. For both a synthetic arithmetic task and natural language, the authors provide evidence that these clusters correspond to human-interpretable model "skills".

**Strengths:**

- To my knowledge, the proposed POLCA loss decomposition is novel.
- The exposition of POLCA (Sec. 3) is clear and well-motivated. I appreciate the use of colors to indicate data points vs basis vectors in the equations.
- The authors provide evidence that datapoints clustered by POLCA trajectories correspond to human-interpretable features or "skills".

**Weaknesses:**

- There are various hyperparameters introduced here (number of Hessian basis vectors to obtain at each iteration, number of training steps between iterations, choice of clustering method and whatever hyperparameters are associated with that clustering method, eg number of clusters) I would appreciate more discussion of how the hyperparameters settings were chosen for the experiments in the paper and robustness of the results to these settings.
- The process of associating an interpretation to data points corresponding to each cluster, per basis vector, per output token, seems to be labor-intensive and possibly error-prone.
- For natural language, POLCA is not compared with any baselines. (For the addition task, there is a comparison with clustering solely on per-token loss, which is good.) Another natural baseline to compare against is with randomly sampled basis vectors, instead of eigenvectors of the loss Hessian.
- Although it is interesting that the POLCA trajectory clusters correspond to human-interpretable skills (at least for the examples given), it is not clear how this interpretation is useful in practice.
	- The paper says, "POLCA analysis could also be combined with model editing methods to better remove abilities learned by the model." I would be interested to see more elaboration on how this could be done + some preliminary results in this direction. For example: does ablating a POLCA basis vector destroy model performance on specifically the skills associated to corresponding clusters?
- Causal experiments like the example in the above point ^ would also help to increase confidence in the current results, which are all observational.

**Questions:**

- It seems that the construction of $B$ depends on the order of iteration over $t=1,\ldots, T$, since the Hessian at checkpoint $t$ is first projected onto the nullspace of $B_{t-1}$. Is there an intuitive reason why you iterate in order of increasing $t$? Are the results robust to reorderings of the iteration?
- It would be helpful to add more details and explanation to Appendix A; currently it is difficult to follow.
	- What is the definition of $g_t(X)$? From Eq. (8), I'm guessing it is the gradient $g_t(X) = \nabla_{\theta_t} L(X; \theta_t)$. But in Eqs. (12-15), it is then used in the denominator of a fraction, so it must be a scalar?
- In general, computing the entire Hessian for large models is prohibitively expensive. Is it any easier if you only need the top $k$ eigenvalues orthogonal to a given subspace?
- If I understand correctly, the explanation given in Sec. 3.1.2 elides over the fact that each datapoint $x\in X$ is a sequence, and that the loss $L$ is further decomposed as a sum over position in the sequence. So, for each basis vector  $b\in B$, if there are $n$ validation sequences each of size $m$, there are $nm$ total POLCA trajectories to cluster. Although, before the clustering is performed, it seems each of the $nm$ trajectories are first grouped by the (ground truth?) _output_ token value. The motivation for grouping by output token is unclear to me; what happens if you don't group by output token?
- It would be interesting to see results on language models larger than the 2-layer model used in Sec. 4. There are various open-source models that provide training checkpoints, e.g. Pythia (Biderman et al. 2023). Is POLCA able to scale to language models on this scale with, say, on the order of 1B parameters?
- What is the parameter count of the model used in the paper's experiments? How many tokens, iterations, epochs was it trained on? What were the training hyperparameters used? What cross-entropy loss did the model attain?
- My understanding is that basis vectors were computed using a subsample of the training set, but clustering was performed on the validation set. Was it checked that the interpretations generalize to points outside those used to find the cluster centroids?

Minor typos:
- Line 130: $\theta$ should be $\theta_t$
- Equation (4) is only strictly true when $B$ is full rank; otherwise it is only an approximation.

---

### Official Review · Reviewer_89LS · 2024-10-26

**Soundness:** 2
**Presentation:** 2
**Contribution:** 2
**Rating:** 5
**Confidence:** 4

**Summary:**

This paper proposes POCLA, a method for decomposing changes in loss along an arbitrary basis of the low rank training subspace. The method is also deployed in synthetic arithmetic and natural language processing.

**Strengths:**

The paper introduces Projection Oriented Loss Change Allocation (POLCA), which decomposes loss trajectories in the model’s parameter space, providing a way to identify hidden conceptual breakthroughs that are often masked in aggregated loss curves.

**Weaknesses:**

1.	While POLCA is shown to perform well, the paper lacks a robust comparison with existing methods for interpretability or skill recovery in LLM training, such as traditional reinforcement learning or trajectory clustering methods. A comparative study could have strengthened the claims about POLCA's uniqueness and efficiency.
2.	While the method is demonstrated on a smaller transformer and Wikipedia dataset, the paper lacks discussion on how POLCA would scale to larger LLMs. Given the computational overhead of Hessian-based calculations, discussing the feasibility of POLCA in large-scale training scenarios would be valuable.
3.	Although POLCA is a promising approach, the paper could benefit from a more comprehensive discussion of its limitations. For example, the reliance on Hessian eigenvectors might introduce noise or dependencies in high-dimensional parameter spaces. Addressing these potential issues would strengthen the claims.

**Questions:**

Please see the weakness section.

---

### Official Review · Reviewer_sts6 · 2024-11-01

**Soundness:** 3
**Presentation:** 3
**Contribution:** 3
**Rating:** 8
**Confidence:** 3

**Summary:**

This paper examine hidden loss breakthroughs in subsets of training examples by decomposing loss changes into a set of basis. The paper show cases where the decomposed loss changes correspond subsets of data points with semantically interpretable conceptual similarity. The paper further shows that the proposed method recovers interpretable clusters that represent breakthoughs in certain model capabilities.

**Strengths:**

- The general idea of examining loss changes in a datapoint-level & decomposed basis is enlightening. It unveils fine-grained learning dynamics that are hidden behind the aggregated loss in prior works. I believe the paper is a nice work towards understanding loss breakthroughs and training dynamics of transformer models.

- The proposed method is neat. How the basis are found & the decomposition is performed & the clustering is done are technically sound.

- Experiments answers research questions efficiently. I specifically appreciate Sec. 4.2. where the authors compare the proposed approach to directly recovering concepts from the exact loss.

- The writing and presentation is clear.

**Weaknesses:**

Despite the work is quite solid in its presented setups, I am concerned about the scalability of the approach. This matters because the practical utility of training dynamics research are commonly found in large-scale models, like Xia et al. 2023 the authors cited.

- The experiments are performed on quite small transformer language models (a 2-layer transformer model).

- If I understand correctly, finding the basis (in Sec. 3.1.1) requires computing the Hessian matrix of size $d \times d$, which is infeasible  even for a relatively "small" LLM with 1B parameter, because the Hessian has the same cost as storing 1 billion 1B models. Is it the main reason why the authors only performed experiments on small transformers?

- If so, I suggest the authors to discuss the scalability issue in the Limitations section, and consider viable workarounds like approximations.

- I think the computational efficiency of the entire method is bottlenecked by the computation cost of Hessian matrix in Sec. 3.1.1, as loss decomposition along the basis (Sec. 3.1.2) & clustering are quite computationally efficient. Could you provide more justification about the choice of using the Hessian in Sec. 3.1.1?

For me, the scalability does not quite block the paper from acceptance, but this is certainly a weakness that should be discussed.

**Questions:**

See weakness.

---

### Official Review · Reviewer_5JQC · 2024-11-02

**Soundness:** 2
**Presentation:** 2
**Contribution:** 2
**Rating:** 5
**Confidence:** 4

**Summary:**

The paper introduces POLCA, a method via which it is possible to decompose the loss of a language model into parts that can be used to analyse the knowledge obtained by the model in a more detailed fashion. More specifically, the authors show that decomposing the loss via POLCA can be used to analyze the distinct skills obtained by the model during training.

**Strengths:**

- The topic of the paper is of interest to the ICLR community. The authors aim to tackle the problem of understanding the knowledge obtained by a language model, a task that is as of now unsolved. The method proposed by the authors has the goal of making this task more manageable - by creating distinct clusters of samples, it is more feasible to analyze them after training and understand to which skill they correspond. Moreover, the method proposed by the authors is novel, as far as I am aware.

- The method is also technically sound, and appears to result in explainable clusters in the experimental settings examined by the authors.

**Weaknesses:**

- Overall, I think the presentation of the paper can be improved:

  - Section 4 is pretty opaque - it took me several tries to understand the experimental setting described by the authors, mostly due to not immediately understanding what the POLCA vectors being clustered represent and how many skills the model was being examined for. I believe having a separate algorithm table describing the entire process, as well as explaining more succinctly what the precise skills that are being evaluated here are would help.

  - I'm having trouble understanding the point that the Figures in Section 4 aim to convey. These figures contain the main experimental result of this Section, and according to the authors show that clustering using the POLCA vectors demonstrates the skills being learned by the model (Figure 2 in particular). However, this is not immediately clear to me, mainly because a) the authors label the bars in the barplot with some of the skills of the model, but don't actually explain how this label is derived, and b) in Figures 2c and 2i a significant portion of the data points are labeled as outliers by the clustering algorithm. I would be grateful if the authors could elaborate on these points.

- I also find that the experimental portion of the paper is not convincing enough:

  - In Section 4 (experiment on synthetic data), as mentioned above only clusters for select POLCA trajectory vectors are presented, with a significant portion being labeled as outliers. The additional settings presented in the appendix for the same experiment show that in some cases, more vectors will end up being labeled as outliers than being part of actual clusters. This undermines the usefulness of the method in my opinion. I believe that some statistics on how often clustering is "succesful" (in the sense that the algorithm returns useful clusters) would elucidate this point (such as what is the average percentage of points being marked as outliers when clustering).

  - In Section 5, the experiment concluded is mostly qualitative, and focuses only on two tokens. While I understand that analysing every single output token is not practical, the analysis performed here is quite basic, and not particularly convincing. More specifically, it is not clear how distinct these skills are (e.g. some contexts of the token "and" have overlap between the "List" and "Token after comma" skills). It is not explicitly stated how many elements are being used to decide which skill corresponds to each cluster (the authors quantify it as a proportion of the elements in the cluster, but the fact that the clustering can have a large amount of outliers as seen in the previous sections makes the absolute amount unclear). I would be grateful if the authors could clarify this.

Overall, while I believe the method is interesting and shows promise, the results presented in this paper don't appear as convincing.

**Questions:**

In addition to the points I made above, I would be grateful if the authors could also let me know whether calculating the loss for a specific token is done across all instances of the token (irrespective of past context), or whether POLCA takes into consideration the past context somehow.

---

### Official Review · Reviewer_ZB6K · 2024-11-03

**Soundness:** 2
**Presentation:** 2
**Contribution:** 1
**Rating:** 3
**Confidence:** 4

**Summary:**

The paper proposes Projection Oriented Loss Change Allocation (POLCA) to identify hidden conceptual "breakthroughs'' during training, which are otherwise obscured in traditional aggregate loss curves. By decomposing loss along specific subspace directions and over individual examples, the authors aim to reveal clusters of datapoints that have similar training trajectories, highlighting "breakthroughs" in the model's learning process. They validate POLCA on synthetic arithmetic and natural language datasets.

**Strengths:**

- The problem is interesting. Identifying hidden dynamics in training is an intriguing area that deserves attention.
- The proposed method of clustering data based on loss changes along specific directions is sensible, potentially offering new perspectives on interpretability if further validated.

**Weaknesses:**

1. **Insufficient Discussion of Related Work and Ambiguous Terminology**

   The paper does not sufficiently position POLCA within the broader context of related work, missing key opportunities to discuss previous methods in phase transition analysis, loss dynamics, and interpretability—even though interpretability is a central claimed benefit. This gap makes it difficult to assess the novelty and significance of POLCA. Although the background section lists some relevant points, these feel more like brief justifications than a thoughtful review of prior literature. Consequently, the section reads more like a positioning statement than a critical examination of foundational work.

   Additionally, the paper introduces numerous terms without adequate definitions, which creates challenges in readability and comprehension. Terms such as “breakthroughs,” “shifts,” “structures,” “breakthrough elisions,” “conceptually meaningful,” “loss Hessian,” “base,” “basis,” and “skill” are used without prior explanation, making it hard for readers to fully grasp POLCA’s approach and its connections to existing research. Providing clear definitions for these terms would greatly enhance the paper’s clarity and accessibility.

2. **Unnecessary Complexity in Methodology**

   POLCA’s reliance on second-order Hessian approximations appears overly complex, particularly when simpler methods could likely achieve similar insights. For example, if the main goal is to cluster training trajectories for individual samples, established methods like saliency maps, activation pattern analysis, or influence functions might provide a more straightforward and interpretable alternative. While influence functions also involve Hessian computations, they are a widely recognized technique and may offer a more practical and accessible alternative to POLCA’s decomposition approach.

3. **Inconsistent and Confusing Notation in Algorithm Formulation**

   The mathematical notation in POLCA’s formulation is inconsistent and unclear in several places, which hampers the paper’s readability. For instance, the symbol $\phi$ in line 127 is ambiguous—if it represents an empty set, it is unclear how an inverse of an empty set is computed in line 128. Notational inconsistencies, such as $\mathcal{L}$ in line 130 versus $L$ in line 148, are also left unexplained, making the flow of the loss formula challenging to follow. Additionally, symbols like $\perp$ (presumably indicating orthogonality) are not defined. If $\perp$ is meant to denote orthogonality of $\Pi$ to the matrix $\mathcal{H}$, this should be clarified, along with citing the method (e.g., Lanczos method) used to find eigenvectors, as referenced in line 131.

4. **Insufficient Experimental Validation**

   The experimental setup appears limited and lacks rigor, especially in the natural language setting, which raises questions about the practical utility of POLCA. For instance, the choice of a two-layer transformer model with an embedding size of 512 is not justified, nor is there any variation in model configurations. Why were these specific parameters chosen, and would POLCA yield similar results with different configurations, such as deeper layers or alternate embedding sizes? The limited variation in hyperparameters weakens confidence in the generalizability of the findings. Additionally, while POLCA seems versatile, it is tested on only language modeling task; applying it to more diverse tasks would have better demonstrated its applicability. Finally, the absence of baseline （see 2）comparisons further undermines confidence in the purported advantages of POLCA.

5. **Unclear Practical Relevance**

   The practical relevance of detecting these “hidden breakthroughs” is not well-argued, and without concrete examples, it remains unclear how this information would enhance training efficiency, model interpretability, or overall performance. Strengthening the argument for the real-world utility of POLCA would make its contributions to applied machine learning more compelling.

**Questions:**

- How does POLCA perform compared to simpler methods that cluster datapoints based solely on their loss trajectories, without the eigenvector-based decomposition? For example clustering datapoints based on their saliency maps，influence functions， activation patterns etc
- What are the computational costs associated with POLCA, and how does it scale to larger datasets or deeper models? How does it relate  to the specific eigen decomposition method used?
- Could the authors provide concrete examples of how identifying these "hidden breakthroughs" translates to improved model performance or training insights in real-world tasks? Any experimental supports on this would be great for convincing readers the applicability of the proposed method.

---

### Meta-Review · Area_Chair_NwHc · 2024-12-23

**Metareview:**

Summary: The paper introduces POLCA (Projection Oriented Loss Change Allocation), a method for identifying hidden conceptual breakthroughs during language model training. By decomposing the loss changes along specific subspace directions, POLCA uncovers clusters of datapoints with synchronized training trajectories, revealing underlying skills or concepts learned by the model. Empirical studies are conducted on synthetic arithmetic tasks and natural language datasets to show that the method recovers some meaningful conceptual clusters that are obscured by traditional loss aggregation.

Strengths:

Weakness:
- Lukewarm response from all but one reviewer and the positive reviewer didn't champion the paper
- Limited experimental validation: Only tested on small models (2-layer transformers) and restricted set of tasks
- Lack of comprehensive comparisons to baselines
- Scalability concerns: even hessian-vector pretty expensive for 1B+ models
- Practical utility: Labor-intensive interpretation process and limited discussion of practical applications (as suggested by one of reviewer: POLCA + Model editing can be very interesting)
- Many hyperparameters without thorough ablation studies
- Some unclear design choices in clustering approach

Decision:
Decision: While the work shows promise and addresses an important problem, unfortunately, the paper can't be accepted in its current form and addressing all the concerns would warrant another round of reviewing.

**Additional Comments On Reviewer Discussion:**

We thank the authors for engaging during the discussion phase towards improving the paper. Below are some of the highlights:

1. Scalability:
- Clarified use of Hessian-vector products instead of full Hessian
- Added discussion of alternative basis construction methods
- Still acknowledged limitations with larger models

2. Experimental Validation:
- Added per-token loss clustering baseline for language tasks
- Included additional analysis of basis vector choices
- Explained model size choice based on prior work

3. Hyperparameters:
- Added justification for parameter choices
- Included more details about clustering approach
- Expanded experimental details in appendix

4. Practical Utility:
- Better positioned work as primarily advancing scientific understanding
- Acknowledged limitations of manual interpretation
- Suggested potential automated approaches like use in model editing

In summary, though the authors provided good clarifications in rebuttalbut concerns still remain and unfortunately reviewers didn't engage much. Many of the suggested improvements, however, would require substantial revision beyond what's possible in the rebuttal period.

---

### Decision · Program_Chairs · 2025-01-22

Reject